# Normalized Indices Derived from Visceral Adipose Mass Assessed by Magnetic Resonance Imaging and Their Correlation with Markers for Insulin Resistance and Prediabetes

**DOI:** 10.3390/nu12072064

**Published:** 2020-07-11

**Authors:** Jürgen Machann, Norbert Stefan, Robert Wagner, Andreas Fritsche, Jimmy D. Bell, Brandon Whitcher, Hans-Ulrich Häring, Andreas L. Birkenfeld, Konstantin Nikolaou, Fritz Schick, E. Louise Thomas

**Affiliations:** 1Institute for Diabetes Research and Metabolic Diseases (IDM) of the Helmholtz Center Munich at the University of Tübingen, 72076 Tübingen, Germany; norbert.stefan@med.uni-tuebingen.de (N.S.); Robert.wagner@med.uni-tuebingen.de (R.W.); andreas.fritsche@med.uni-tuebingen.de (A.F.); andreas.birkenfeld@med.uni-tuebingen.de (A.L.B.); fritz.schick@med.uni-tuebingen.de (F.S.); 2German Center for Diabetes Research (DZD), 72076 Tübingen, Germany; haerinhu@icloud.com; 3Section on Experimental Radiology, Department of Diagnostic and Interventional Radiology, University Hospital Tübingen, Hoppe-Seyler-Str. 3, 72076 Tübingen, Germany; 4Department of Endocrinology, Diabetology and Nephrology, University Hospital Tübingen, 72076 Tübingen, Germany; 5Research Centre for Optimal Health, School of Life Sciences, University of Westminster, London W1W 6UW, UK; J.Bell@westminster.ac.uk (J.D.B.); B.Whitcher@westminster.ac.uk (B.W.); L.Thomas3@westminster.ac.uk (E.L.T); 6Department of Diagnostic and Interventional Radiology, University Hospital Tübingen, 72076 Tübingen, Germany; konstantin.nikolaou@med.uni-tuebingen.de

**Keywords:** visceral adipose tissue, normative values, insulin resistance, prediabetes, magnetic resonance imaging, visceral fat index

## Abstract

Visceral adipose tissue (VAT) plays an important role in the pathogenesis of insulin resistance (IR), prediabetes and type 2 diabetes. However, VAT volume alone might not be the best marker for insulin resistance and prediabetes or diabetes, as a given VAT volume may impact differently on these metabolic traits based on body height, gender, age and ethnicity. In a cohort of 1295 subjects from the Tübingen Diabetes Family Study (TDFS) and in 9978 subjects from the UK Biobank (UKBB) undergoing magnetic resonance imaging for quantification of VAT volume, total adipose tissue (TAT) in the TDFS, total abdominal adipose tissue (TAAT) in the UKBB, and total lean tissue (TLT), VAT volume and several VAT-indices were investigated for their relationships with insulin resistance and glycemic traits. VAT-related indices were calculated by correcting for body height (VAT/m:VAT/body height; VAT/m^2^:VAT/(body height)^2^, and VAT/m^3^:VAT/(body height)^3^), TAT (%VAT), TLT (VAT/TLT) and weight (VAT/WEI), with closest equivalents used within the UKBB dataset. Prognostic values of VAT and VAT-related indices for insulin sensitivity, HbA1c levels and prediabetes/diabetes were analyzed for males and females. Males had higher VAT volume and VAT-related indices than females in both cohorts (*p* < 0.0001) and VAT volume has shown to be a stronger determinant for insulin sensitivity than anthropometric variables. Among the parameters uncorrected VAT and derived indices, VAT/m^3^ most strongly correlated negatively with insulin sensitivity and positively with HbA1c levels and prediabetes/diabetes in the TDFS (R^2^ = 0.375/0.305 for females/males for insulin sensitivity, 0.178/0.148 for HbA1c levels vs., e.g., 0.355/0.293 and 0.144/0.133 for VAT, respectively) and positively with HbA1c (R^2^ = 0.046/0.042) in the UKBB for females and males. Furthermore, VAT/m^3^ was found to be a significantly better determinant of insulin resistance or prediabetes than uncorrected VAT volume (*p* < 0.001/0.019 for females/males regarding insulin sensitivity, *p* < 0.001/< 0.001 for females/males regarding HbA1c). Evaluation of several indices derived from VAT volume identified VAT/m^3^ to correlate most strongly with insulin sensitivity and glucose metabolism. Thus, VAT/m^3^ appears to provide better indications of metabolic characteristics (insulin sensitivity and pre-diabetes/diabetes) than VAT volume alone.

## 1. Introduction

The prevalence of obesity (defined by a body mass index (BMI) greater than 30 kg/m^2^) is continuously increasing in the western world [1], with parallel increases in resource-poor nations [2] and even in the Arctic region [3]. There are predictions that nearly 100% of Americans will suffer from overweight or obesity by 2050 [1] while others forecast that this epidemic will eventually plateau [4,5]. It is well accepted that obesity is a major risk factor for metabolic syndrome, insulin resistance, type 2 diabetes and concomitant disorders, including cardiovascular diseases and stroke [6,7,8,9]. Abdominal obesity in particular is reported to be an important predictor of metabolic diseases and is considered a key feature of the metabolic syndrome [10,11,12].

Many epidemiological studies have aimed at better understanding of the contribution of body composition to the development of metabolic diseases, especially in large cross-sectional populations. These studies did not often directly measure body fat content; instead they relied on indirect proxies such as body mass index (BMI) for total body fat and waist circumference or waist-to-hip ratio (WHR) for visceral adipose tissue (VAT). It is well established that BMI is associated with a higher risk of cardiovascular diseases or type 2 diabetes. In addition, adults with a higher waist circumference have an increased risk of cardiometabolic disease, compared to adults with a lower waist circumference [13,14]. However, these proxy measures of adiposity do not reflect the metabolic risk in an appropriate manner, particularly as they fail to capture many important obesity phenotypes. This is particularly the case in subjects with a high BMI (>30 kg/m^2^) but a “metabolically-healthy obesity” phenotype, who are characterized by relatively low volumes of VAT, despite significant levels of adipose tissue elsewhere in the body or significant levels of musculature. Most individuals characterized by this phenotype have insulin sensitivity within the normal range [15]. Conversely, subjects with the “thin outside, fat inside” phenotype, are characterized by elevated VAT, despite a BMI and WHR being within the normal range. This phenotype is associated with an elevated risk of developing metabolic disease [16].

Magnetic resonance imaging (MRI) enables reliable quantification of volumes of different adipose and lean tissue compartments, using suitable acquisition techniques [17,18,19] and post processing approaches [19,20,21] for body profiling. As such MRI can be used for direct measurements of adiposity in cross-sectional [22,23,24] and interventional studies. In the latter, a reduction of VAT was found to associate with improvement in insulin sensitivity in cohorts of subjects having an increased risk of diabetes [25,26].

However, there is a lack of consistency regarding the most appropriate measurement of adiposity by MRI. Many studies use single slice area measurement of VAT at a specified anatomical landmark within the abdomen, typically L4/L5. The most in-depth and accurate method for assessing VAT involves measuring the entire abdomen and quantifying total VAT volume. This is subsequently presented as a biomarker within metabolic related research as total volume of VAT in liters, typically without normalization/correction for size (especially height), gender or age. However, it is evident that the cardiometabolic risk attributed to a specific VAT volume may differ according to body habitus, i.e., a large subject (e.g., a tall male with a height of 1.90 m) may have a very different metabolic impact compared to the same VAT volume within a small subject (e.g., a small female with a height of 1.50 m). Recently, several studies have proposed applying a VAT index, using height squared as normalization factor, which is comparable to the calculation of BMI [27,28,29]. However, there is no evidence that this normalized approach is a more appropriate measure to better characterize the risk of insulin resistance and/or impaired glucose metabolism, e.g., determined by impaired fasting glucose (IFG), impaired glucose tolerance (IGT) and/or increased HbA_1c_, compared to the absolute VAT volume.

Therefore, the basic overarching hypothesis of this study in two large MRI-based cohort studies—i.e., the Tübingen Diabetes Family Study (TDFS) and the UK Biobank (UKBB)—is that a normalized VAT index, considering individual body constitution, shows a better correlation to markers of insulin resistance and impaired glucose metabolism than absolute VAT volume. Furthermore, the benefit of using direct VAT measurements for indication of possible insulin resistance and impaired glucose metabolism, rather than relying on indirect anthropometric measures such as waist and hip circumference (as an approximate for adipose tissue distribution), BMI (as a measure of total body fat mass), age and/or gender, is being tested. Finally, we have attempted to determine which VAT index has the best diagnostic performance for prognostication of insulin resistance and prediabetes/diabetes. To determine this, we have investigated several different VAT-related indices, normalized for height taken as our key marker of body habitus, since height is known to be a protective factor, as taller individuals are less susceptible to metabolic and cardiovascular diseases [30]. We have also explored the impact of other potential factors including total adiposity, muscle and body weight on VAT indices given their established relationship with incidence of insulin resistance and impaired glucose metabolism. Since it is well established that females have significantly lower levels of VAT, compared with males of a similar BMI [27,28,29], we performed analyses separately for females and males.

## 2. Materials and Methods

TDFS Cohort: The TDFS is an ongoing study that predominantly recruits participants from Southern Germany aged >18 years old and having an increased risk of diabetes [31]. This is achieved by targeting family members of patients having type 2 diabetes and by announcements about new studies via institutional websites and flyers. Subjects are deemed at increased risk for metabolic diseases if they have overweight (BMI > 27kg/m^2^), are first degree relative of a subject with type 2 diabetes, have impaired glucose tolerance and/or had gestational diabetes during pregnancy. In total, 1295 subjects (801 females, 494 males, mean age 44.7 years), who underwent MRI measurements to quantify fat mass, were included in this cross-sectional retrospective analysis. All volunteers gave written informed consent and the examinations were approved by the local Ethics Committee.

All examinations were performed in the early morning after overnight fasting in the facilities of the University Hospital Tübingen.

UKBB Cohort: 9978 subjects (5186 females, 4792 males, mean age 55.4 years) were included in this cross-sectional analysis. The UKBB (see www.ukbiobank.ac.uk for more information) is a large population-based cohort that includes 503,325 individuals aged 40–70 years old, aimed at improving the prevention, diagnosis, and treatment of a wide range of serious and life-threatening illnesses. Baseline assessment included extensive information, via physical measurements, questionnaires, samples, and consent to access medical records. Participants were recruited across the United Kingdom over a five year period beginning in 2006. Subsequent to the initial assessment, 100,000 participants are being recalled for an imaging study including the brain, heart, bones, carotid arteries, and body composition [32]. The study was approved by the North West Multicenter Research Ethics Committee in the United Kingdom. Written informed consent was obtained prior to study entry. The research included in this paper has been conducted using the UKBB resource, project ID 23889. All data is available to bona fide researchers via application to the UKBB.

Exclusion criteria for participation in both studies were general contra-indications for MRI (e.g., cardiac pacemaker, ferromagnetic implants, claustrophobia and pregnancy). Besides these, subjects weighing more than 170 kg (load limit for the patient table) or who could not fit in the bore of the scanner could not be included which may lead to a slight bias in selection since body composition of people having extreme obesity cannot be considered.

### 2.1. Magnetic Resonance Imaging

TDFS Cohort: MRI examinations were performed on a 1.5 T whole body imager (Magnetom Sonata, Siemens Healthineers, Erlangen, Germany). Subjects lay in prone position with extended arms for assessment of the whole body, taking about 20 min in total. For radiofrequency (RF) irradiation and signal acquisition, the body coil was used. A T1-weighted fast spin-echo sequence with echo train length 7 was applied with following measurement parameters: echo time (TE)/repetition time (TR) = 11/490 ms, slice thickness 10 mm, gap between slices 10 mm, field of view: 480–560 mm depending on the breadth of the subject, matrix: 256 × 178, 5 slices per sequence, acquisition time 13 s, allowing breath-hold acquisition in abdominal region. One rearrangement of the subject was necessary due to the limited table shift. Examinations were performed from iliac crest to feet and from iliac crest to fingers. Post processing was conducted applying an automatic segmentation algorithm based on Matlab (The MathWorks, version 7.5.0) as described in [21]. Areas of adipose tissue (AT) and lean tissue (LT) were automatically calculated, and separation of visceral and other adipose tissue depots was automatically performed between femoral head and thoracic diaphragm using an extended snake algorithm [21]. Figure 1 shows a typical segmentation in a 48-year-old male subject with BMI = 28.1 kg/m^2^.

Slight imperfections of segmentation caused by inhomogeneity of the receiver coil were manually corrected, after visual inspection of the resulting dataset enabling an individualized whole-body fat and lean tissue profile to be generated. Tissue volumes in the 10 mm gaps between slices were linearly interpolated, enabling quantitative assessment of the volumes of VAT, total body adipose tissue (TAT), and total body lean tissue (TLT).

Of note, yellow bone marrow from the entire skeleton and—depending on the fat/water ratio—red bone marrow was included in the analysis. However, as this contributes a very similar amount for subjects with the same height, it only moderately influences the results. Segmentation of tissue compartments could be successfully performed in all participants, with user independent automatic segmentation taking 2:30–3 min, this manual correction following visual inspection (if necessary) requiring an additional 3–5 min per subject.

UKBB: Subjects were scanned in a Siemens MAGNETOM Aera 1.5 T MRI scanner (Siemens Healthineers, Erlangen, Germany) using a 6-min dual-echo Dixon Vibe protocol, providing a water and fat separated volumetric data set covering neck to knees as previously described [33]. Body composition analyses were performed using AMRA Profiler Research (AMRA Medical AB, Linköping, Sweden), imaging derived outputs included VAT and abdominal subcutaneous adipose tissue (ASAT), and thigh muscle volume. VAT was defined as the adipose tissue within the abdominal cavity, excluding adipose tissue outside the abdominal skeletal muscles and adipose tissue and lipids within and posterior to the spine and posterior to the back muscles. ASAT was defined as subcutaneous adipose tissue in the abdomen from the top of the femoral head to the top of the thoracic vertebrae T9 as shown in a coronal view of a male subject of the UKBB in Figure 2.

The following VAT-dependent indices were calculated based on VAT volume:
●VAT/mVAT/body height[L/m]●VAT/m^2^VAT/body height^2^[L/m^2^]●VAT/m^3^VAT/body height^3^[L/m^3^]●%VATVAT/total adipose tissue[%] *●VAT/TLTVAT/total lean tissue[%] *●VAT/WEIVAT/body weight[L/kg]

* The UK Biobank (UKBB) MRI protocol does not include measurement of total adipose or lean tissue, therefore within this dataset %VAT corresponds to VAT/total abdominal adipose tissue and VAT/TLT (total lean tissue) corresponds to VAT/thigh muscle volume.

### 2.2. Anthropometric Parameters and Metabolic Measurements

TDFS Cohort: Immediately after the MRI examination, subjects were transferred to our metabolic ward. Body height in standing position and body weight by the nearest 0.1 kg were measured, and BMI was calculated as kg/m^2^. Waist circumference (measured at the midpoint between the lower margin of the last palpable rib and the top of the iliac crest) and hip circumference (around the widest portion of the buttocks) were measured using a stretch-resistant tape, and waist-to-hip ratio (WHR) calculated. For determination of insulin sensitivity, all subjects underwent a 75-g oral glucose tolerance test and venous plasma samples were obtained at 0, 30, 60, 90, and 120 min for determination of plasma glucose and insulin. According to Matsuda and DeFronzo [34], insulin sensitivity was expressed as insulin sensitivity index (ISI_Mats_) in arbitrary units. In order to identify subjects with disturbed glucose metabolism—i.e., IFG, IGT or newly diagnosed diabetes—glucose levels at baseline (Gluc_0_) and after 2 h (Gluc_120_) of the oral glucose tolerance test were considered. Herein, 100 mg/dL < Gluc_0_ < 126 mg/dL indicates IFG and Gluc_120_ > 140 mg/dL IGT. Additionally, HbA_1c_ was determined to identify people with persistently elevated blood glucose (HbA_1c_ > 5.7%, or 38.8 mmol/mol).

UKBB Cohort: The metabolic measurements included in the TDFS such as oral glucose tolerance test are not available in the UKBB cohort, therefore HbA_1c_ was used to identify a comparative cohort with persistently elevated blood glucose (HbA_1c_ > 5.7%, or 38.8 mmol/mol).

### 2.3. Statistical Analyses

All calculations in the TDFS Cohort were performed with JMP (JMP^®^ 13.0.0 SAS Institute, Cary, NC, USA), and within the UKBB dataset using RStudio (RStudio, Inc., Boston, MA, USA). Data are reported as mean ± SD unless otherwise stated. Distribution of parameters was tested for normality using Shapiro-Wilk W test. Non-normally distributed parameters were log-transformed to approximate normal distribution before statistical analyses. A two-sided unpaired Student t test was used to test for gender-related differences. Univariate linear correlation analyses were used to analyze the coefficient of determination (R^2^) between ISI_Mats_/HbA1c and VAT-related indices. *p*-values < 0.05 were considered statistically significant. Regarding insulin sensitivity, the lowest quartile of ISI_Mats_ was used as threshold for categorization of insulin resistant (IR, ISI_Mats_ < 5.25 a.u.) and insulin sensitive (IS, ISI_Mats_ ≥ 5.25 a.u.) subjects. Second classification was done regarding glucose metabolism, prediabetes/diabetes (subjects having one of the criteria IFG, IGT, increased HbA1c or diabetes) and healthy subjects being in the normal range for all three parameters. In order to test whether VAT is a superior determinant of insulin sensitivity (ISI_Mats_ as continuous variable) and prediabetes (categorized variable) than simpler measures, i.e., age, sex, waist and hip circumferences, WHR and BMI, forward stepwise regression analyses were performed. Whether or not the derived VAT-indices are superior to uncorrected VAT volume in determining insulin sensitivity and prediabetes/diabetes was tested by a Hotelling’s T^2^-test after performing a Fisher z-transformation of the Pearson’s correlation coefficients r.

## 3. Results

MRI derived values of body composition, anthropometric data, and HbA_1c_ were available for subjects in both cohorts, while the additional parameters Gluc_0_, Gluc_120_, and ISI_Mats_ were only available for subjects in the TDFS Cohort. MRI data of the TDFS were investigated in more detail: Appendix A depicts the correlation between MRI-derived total body volume, corrected for density of adipose tissue (0.91 kg/L) and bone mass, by the formula proposed by Heymsfield [35], and body weight in kg determined on a scale immediately after the MRI examination, with an excellent agreement.

First, we tested in the TDFS whether MR-measured VAT is superior to the measurement of common anthropometric variables in determining insulin resistance and prediabetes. In stepwise multiple regression analyses high VAT volumes are superior to high BMI, hip/waist circumference, and WHR in determining both low insulin sensitivity and prediabetes (Table 1).

Next, we tested the relationship of VAT and insulin sensitivity adjusted for age and gender. There is no interaction between age and VAT (p for interaction of age*VAT = 0.1519), thus in the following analyses age is included as a continuous variable.

### 3.1. Gender Related Characteristics of Subjects in the TDFS and UKBB

Anthropometric and metabolic data of the entire study cohort (as well as separated by gender) are listed in Appendix A. Generally, males were older and characterized by a lower insulin sensitivity compared to females. In the TDFS, 372/801 females (46.4%) and 215/494 males (43.5%) had a BMI of 30 kg/m^2^ or higher, thus classified as having obesity. In contrast, the prevalence of obesity in the UKBB was much lower with only 22.9% females and 24.3% males classified as having obesity. In both cohorts, males had significantly higher WHR than females.

The total volumes of tissue compartments and the VAT-related indices are given in Appendix A. In both studies, males had significantly higher lean tissue mass compared to females, whereas females were characterized by significantly higher TAT mass in the TDFS (whole-body MRI) but not in the UKBB (abdominal MRI). In contrast, at comparable BMI, VAT was found to be almost twice as high in males compared to females in both cohorts (VAT-ratio males vs. females = 1.90/1.89 in TDFS/UKBB). These differences remain significant for all VAT-related indices.

As body composition changes with increasing age [28,36,37], in a more detailed analysis, females and males were divided in age tertiles, (TDFS age group 1: young adults, age 19–37/18–39 yrs for females/males, age group 2: middle-aged adults, age 38–51/40–54 yrs, and age group 3: seniors age 52–77/55–75 yrs.; UKBB age group 1: age 40–51/40–52 yrs for females/males, age group 2: age 52–59/53–60 yrs, and age group 3: age 60–70/63–70 yrs). Differences in age groups are given in Appendix A for TDFS and Appendix A for UKBB. In brief, VAT increased with age for both females and males, showing an increment of up to 101% (age group 3 vs. age group 1 in IS females) in the TDFS. The corresponding age-related increase (age group 3 vs. age group 1 in healthy females) in the UKBB cohort was considerably smaller at 13.6%, reflecting the narrow age range in that data set.

### 3.2. Determinants of Insulin Resistance and Impaired Glucose Metabolism in the TDFS

In order to evaluate which VAT-related index has the best diagnostic performance to prognosticate insulin sensitivity and prediabetes, univariate linear correlation analyses were performed. Compared to VAT volume, coefficient of determination (R^2^) to the continuous variables of ISI_Mats_ and HbA1c systematically increased when correcting for body height (VAT/m), surface (VAT/m^2^) and volume (VAT/m^3^), all of them considering the body height of the subjects in different dimensions, whereas R^2^ is lower for indices comprising TAT, lean tissue or weight. All coefficients of determination of the variables VAT, VAT-derived indices and conventional variables (anthropometrics and sex) are given in Table 2. The marked differences in R^2^ between HbA1c and age and/or VAT-related parameters might be caused by the unequal age range of the two cohorts (i.e., 18–77 for the TDFS and 40–70 for the UKBB). By categorization of the variables—i.e., quartiles of ISI_Mats_ and IFG, IGT, HbA1c as mentioned above—the differences in mean VAT volume and calculated indices were more pronounced for IR vs. IS compared to PRED/DIAB vs. healthy subjects in both females and males. All derived VAT-related indices for the subjects in the TDFS are presented in Appendix A. In order to test whether the diagnostic performance of VAT/m^3^ is significantly better than the uncorrected volume of VAT for prognostication of insulin sensitivity, a one-sided Hotelling’s T^2^-test was performed, considering Fisher z-transformed linear correlation coefficients r between VAT, VAT/m^3^ and ISI_Mats_ (continuous variables). As a result, the diagnostic performance of VAT/m^3^—as compared to VAT—is significantly better in females (t = −5.848, *p* < 0.001) and in males (t = −2.121, *p* = 0.017). Hotelling’s T^2^-test for HbA1c resulted in significantly better performance for VAT/m^3^ in both males and females (t = −9.724/−4.798, *p* < 0.001/0.001 for females/males). Thus, VAT/m^3^ has been shown to have a significant better correlation with markers of insulin sensitivity and prediabetes than uncorrected VAT volume.

### 3.3. Determinants of Impaired Glucose Metabolism in the UKBB

As mentioned earlier, data from the UKBB could only be analyzed in respect of glucose metabolism with HbA1c level, as a marker for chronically elevated blood glucose. Similar to the results found in the TDFS cohort, data from the UKBB cohort showed weak correlations between VAT-related indices and HbA1c levels (0.014 < r^2^ < 0.046) for both males and females. However, as with the TDFS cohort in this cross-sectional observation of the general population, highest R^2^ values were obtained for VAT/m^3^ as shown in Table 2. Derived VAT-related indices for the participants in the UKBB are given in Appendix A.

## 4. Discussion

MR-based phenotyping for body composition profiling is frequently applied in cross-sectional and intervention studies in subjects at increased risk for metabolic diseases, as well as in descriptive studies reflecting the general population without being characterized by having specific diseases and/or risk factors, aiming to determine sub-clinical disease burden in the general population. In this context, increased VAT mass was found to be a main determinant of cardiometabolic risk.

Several established techniques, such as T1-weighted MRI [25] or phase sensitive techniques (e.g., Dixon-based imaging) [18,38], are established for data acquisition, and segmentation of adipose and lean body compartments can reliably be performed applying semiautomatic or fully automated algorithms in short evaluation times. However, data interpretation for a correct assignment of the resulting tissue volumes lacks standardization, as just considering the volume is not necessarily reliable for subjects of differing height, gender, and age. Normalized VAT-derived indices incorporate the subjects’ height in addition to volumes of total adipose tissue (TAT) or total lean tissue (TLT). It was investigated as to whether there was an added benefit of using these indices to better characterize the risk of insulin resistance and prediabetes (i.e., impaired glucose metabolism) than simply using directly measured VAT-volume in two separate cohorts. From our analyses it was established that a volumetric correction—i.e., dividing the VAT volume by (height)^3^—provided the best discrimination between insulin sensitive and insulin resistant as well as between healthy subjects and subjects having prediabetes/diabetes. Subcutaneous adipose tissue and lean tissue—mainly reflecting muscle mass—might be protective factors in respect of metabolic lapse; however, this is not reflected when correcting VAT for these quantities as shown for VAT/TLT and VAT/WEI in Appendix A. Compared to the other indices, the percentage of VAT (i.e., %VAT = VAT/TAT) was more weakly correlated to ISI_Mats_. An interpretation of this result might be that this measure is specific for the redistribution of AT, which has previously been observed to change with increasing age as shown in [27,31] and confirmed in this larger study. The increase in %VAT in older women is striking and might be related to postmenopausal hormonal alterations beyond the age of 50, which is commonly associated with the onset of menopause in Caucasian women [39].

Determination of specific cut-off values for VAT or VAT-derived indices for discrimination of insulin resistance and/or prediabetes remains somewhat doubtful since VAT mass alone is not solely responsible for the metabolic status of an individual. Various hormonal and genetic variables [40,41] as well as intrahepatic lipids (i.e., accumulation of fat in the liver) [14,15,25,29,38,42,43,44,45] should also be accounted for.

The correlation between VAT and related derived indices and insulin sensitivity in the TDFS cohort are specific to a cohort of Caucasian subjects at increased risk for metabolic diseases and therefore cannot directly be transferred to the general population. However, the results were replicated in a large population-based epidemiological cohort study (UKBB) for glucose metabolism using HbA1c as a marker for persistently elevated blood glucose. From the results of both cohort studies, TDFS and UKBB, it can be concluded that VAT is not well correlated to impaired glucose metabolism as indicated by elevated HbA1c. The relationship between VAT and insulin sensitivity, as determined by oral glucose tolerance test and given by ISI_Mats_, is more pronounced.

Due to the fact that whole-body MRI is actually being applied in several population-based cohort studies for detailed phenotyping of subjects reflecting the normal population [22,23,29], there will be an increasing applicability for future evaluations.

Limitations of the present study might be the fact that the images in the TDFS study were acquired applying a relatively simple 2D T1-weighted sequence approach. However, this data recording and post-processing is well established and has shown excellent agreement regarding segmentation of spatially extended tissue compartments compared to modern volumetric 3D Dixon-based MRI [46] as applied in the UKBB cohort. This comparability enables the merging of data sets from large cohort studies with very similar mean volumes and ranges of measured adipose tissue, and with similar relationships between the measured variable. The advantage of seamless acquisition with higher spatial resolution is accompanied by the limitation of not acquiring the entire lower extremities, thus hampering calculation of the same indices (e.g., %VAT and VAT/TLT) which is compensated for by using standardized and well-comparable tissue volumes for this purpose.

Another consideration is that normal weight subjects with a BMI < 25 kg/m^2^ are underrepresented in the Tübingen study group, with a paucity of younger subjects in both cohorts, especially the UKBB, where the minimum age was 40 yrs. This reflects the inclusion criteria of both studies and it is feasible that these indices will deviate in children and adolescents due to their inherently different AT distribution [47,48].

## 5. Conclusions

In conclusion, the proposed normative VAT index which volumetrically corrects for body height (VAT/m^3^) has shown to be of slightly but significant better correlation with markers of insulin resistance and prediabetes than simply quantifying the VAT volume, and its use is advised in studies for prospective metabolic research. It has to be determined whether this index is helpful in etiological research and/or even in diagnostic imaging.

## Figures and Tables

**Figure 1 nutrients-12-02064-f001:**
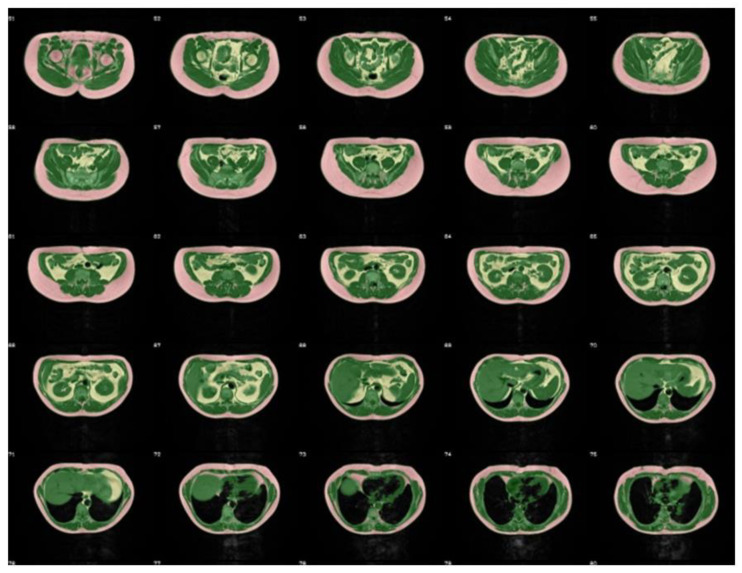
Result for automatic segmentation of lean tissue (green), non-visceral adipose tissue (red) and visceral adipose tissue (VAT) (yellow) between femoral heads and aortic diaphragm in an axial magnetic resonance imaging (MRI) dataset of a 48-year old male subject (BMI 28 kg/m^2^) from the TDFS cohort.

**Figure 2 nutrients-12-02064-f002:**
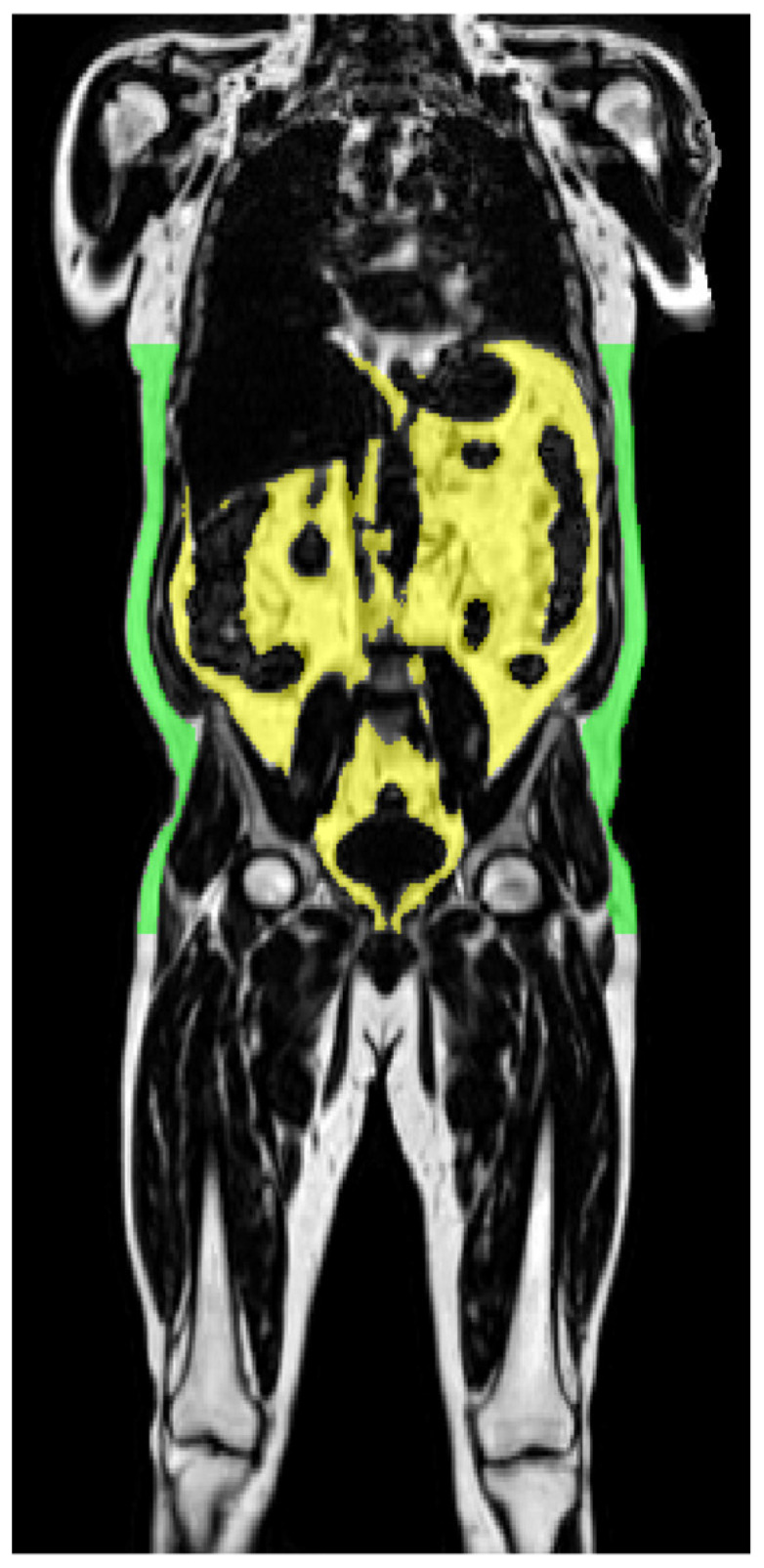
Result for automatic segmentation of abdominal subcutaneous adipose tissue (green) and VAT (yellow) in a coronal view of a 63-year old male subject (BMI 29.4 kg/m^2^) from the UKBB cohort. Posterior thigh muscles were defined as gluteus, iliacus, adductor and hamstring muscles on respective sides and anterior thigh muscles were defined as quadriceps femoris and sartorius [33].

**Table 1 nutrients-12-02064-t001:** a: Determinants of insulin sensitivity (continuous variable) in a stepwise linear regression analysis in the Tübingen Diabetes Family Study (TDFS). b: Determinants of prediabetes (categorized variable) in a stepwise logistic regression analysis in the TDFS.

a			
**Characteristics**	**Estimates**	**F-Ratio**	***p***
log VAT (l)	−0.217	140.837	<0.0001
log BMI	−0.552	42.603	<0.0001
Sex	0.067	51.331	<0.0001
log Hip Circumference	0.518	16.295	<0.0001
log Age	0.083	11.153	0.0009
log WHR	0	1.421	0.2354
log Waist Circumference	0	1.382	0.2400
b			
**Characteristics**	**Estimates**	**Wald/Score ChiSq**	**Prob > Chi-Square**
log age	−2.627	94.072	<0.0001
log VAT (l)	−0.811	10.311	0.0013
Sex	−0.339	14.761	0.0001
log WHR	0	3.004	0.083
log BMI	0	0.083	0.773
log Waist circumference	0	2.662	0.103
log Hip circumference	0	2.612	0.106

**Table 2 nutrients-12-02064-t002:** Coefficient of determination (R^2^) for conventional VAT-related indices and continuous variables of ISI_Mats_ and HbA1c for females and males.

		n	Age [Years]	BMI [kg/m^2^]	WC [cm]	HC [cm]	WHR	VAT [L]	VAT/m [L/m]	VAT/m^2^ [L/m^2^]	VAT/m^3^ [L/m^3^]	%VAT	VAT/TLT	VAT/WEI
**females**														
TDFS	ISI_Mats_	801	0.013	0.274	0.301	0.186	0.159	0.355	0.363	0.369	0.375	0.194	0.349	0.316
TDFS	HbA1c	801	0.235	0.028	0.060	0.028	0.039	0.144	0.155	0.164	0.178	0.151	0.156	0.160
UKBB	HbA1c	4774	0.071	0.023	0.035	0.012	0.035	0.039	0.044	0.045	0.046	0.037	0.042	0.041
**males**														
TDFS	ISI_Mats_	494	0.034	0.288	0.274	0.200	0.123	0.293	0.299	0.302	0.305	0.051	0.267	0.224
TDFS	HbA1c	494	0.254	0.031	0.040	0.016	0.028	0.123	0.133	0.141	0.148	0.107	0.143	0.145
UKBB	HbA1c	4791	0.026	0.026	0.028	0.011	0.028	0.036	0.039	0.041	0.043	0.014	0.043	0.035

WC = waist circumference, HC = hip circumference, WHR = waist-to-hip ratio, VAT = visceral adipose tissue/liters, VAT/m = VAT/body height, VAT/m^2^ = VAT/body height^2^, VAT/m^3^ = VAT/body height^3^, %VAT = VAT/total adipose tissue, VAT/TLT = VAT/total lean tissue and VAT/WEI = VAT/body weight. The UKBB MRI protocol does not include measurement of total adipose or lean tissue, therefore within this dataset %VAT = VAT/total abdominal adipose tissue, VAT/TLT = VAT/thigh muscle volume.

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
