# Peer review of "Normalized Indices Derived from Visceral Adipose Mass Assessed by Magnetic Resonance Imaging and Their Correlation with Markers for Insulin Resistance and Prediabetes"

_nutrients, 2020, doi:10.3390/nu12072064_

Round 1
Reviewer 1 Report
nutrients-830632-tracked
Overall feedback: This is a potentially interesting paper, and it is somewhat improved from the previous version I reviewed, but the analyses of the data appears incomplete and therefore the conclusions are not currently justified by the data.
The abstract would benefit from presentation of more data. For example, it would be helpful to show the data that the correlation for VAT divided by Height squared is better than VAT alone or VAT divided by height.
The paper needs to switch to consistently using person-first language around overweight and obesity throughout, and describing overweight/obesity as something that people have instead of something that they are.
The sentence starting at line 101 does not make sense.
The text at lines 307 to 315 is not clear. For Example, when the authors say that something is improved, what is the comparator?
For the data in table 2, the r squared values seem very similar regardless of whether VAT or vat squared or Vat divided by height etc. is used. I think the authors need to show more evidence of how they showed these r squared values to be different from each other.
The data from table 2 is not easy to follow at all.
The discussion covers a number of aspects that are not relevant to the current work. For example, whether or not VAT can be reliably estimated from a single slice as opposed to the whole abdomen seems irrelevant to the current discussion. Moreover, discussion about automation of VAT assessment also seems irrelevant.
Reviewer 2 Report
I think the authors greatly improved the manuscript by stating the aims of the study more clearly, and performing additional analyses suiting their aims. Overall, the structure and readability of the manuscript improved as well.
However, I still have a few minor comments:
- Abstract: “However, VAT volume alone might not be the best causal variable of insulin resistance and prediabetes or diabetes, as a given VAT volume may differently impact on these metabolic traits based on body height, gender, age and ethnicity.”
I would be careful with mentioning VAT volume as a ‘causal variable’, as I think most research in the field of VAT is observational. The authors mention some intervention studies on lowering MRI-derived indices by lifestyle interventions in the references, however from the introduction it is not clear to me whether these studies showed an effect of lowering VAT on markers of insulin resistance.
- Results: in the Results section, the authors are already interpreting some of their results (for example: “As a result, the diagnostic performance of VAT/m³ is significantly better in females (t = -3.304, p < 0.001) but only shows a non-significant trend in males (t = -1.470, p = 0.071) probably due to the lower number of male subjects included.” I would move sections like these that contain interpretation of the results to the discussion/conclusion section.
- Conclusion (line 381-383): what do the authors mean by ‘epidemiological studies’? They mention cross-sectional studies and intervention studies in the first line, which are both also considered epidemiological studies. Maybe they mean ‘descriptive studies’? This is not clear to me.
Reviewer 3 Report
We thank the authors for going through the effort of updating their manuscript based on the feedback in the first review, however some issues remain.
- The results section starts off with describing the results from stepwise regression analyses that use insulin sensitivity and prediabetes as an outcome. These analyses are only very briefly mentioned at the end of the introduction, but without any details. There is also no (further) information on these analyses in the methods section. As a result, it is not clear how the results in Tables 1a and 1b are arrived at (even with the author’s reply which includes some extra information that is not in the manuscript). If you choose to keep them in the manuscript (see next item), please provide more details.
- A second point concerns the goal, use and interpretation of the stepwise regressions. These analyses were included in response to the first reviews to identify whether VAT is more strongly associated with insulin sensitivity and pre-diabetes than conventional variables/models such as sex or waist circumference. At first glance (but again given the lack of details this is speculative), Tables 1a and 1b represent the results of a forward stepwise regression, where in each step additional variables are added to the model based on statistical grounds and the R2 of the entire model is presented. This does not provide a good basis for the comparison of the association of the different variables with the outcomes of interest as it is now unclear how strongly each factor on their own associates with outcomes. Rather, the current approach gives some idea about how different factors improve total model performance on top of the variables included in the previous steps.
If the authors want to assess and compare the associations of conventional/current approximations they should be analyzed in similar fashion to what is done in the main analysis (with VAT-indices). That is, put each of these (conventional) variables in a model on their own (without including other variables) and compare R2 against VAT and the VAT-indices. For an even more complete analysis, the analyses could be expanded based on the recommendations from the first review round by including currently existing (multivariable) models of insulin resistance/(pre)diabetes. Even a simple model including for example BMI, sex and age will probably already result in a higher performance (if we stick to considering R2 as the main measure for this) than just one individual VAT measurement.
Minor remarks:
- Page 3 of 22, lines 101-105:
“The benefit of using direct VAT measurements for prognostication of insulin resistance and impaired glucose metabolism rather than relying on indirect anthropometric measures such as waist circumference and hip circumference (as an approximate for adipose tissue distribution), BMI (as a measure of total body fat mass), age and/or gender”
Currently, the core of the sentence is:
“The benefit of using direct VAT measurements …. rather than …. “
This is an incomplete sentence that doesn’t come to a stop/conclusion.
- Table 1a:
The table description states it represents results of stepwise logistic regression. Given that insulin sensitivity is continuous this should be linear regression (if you chose to keep these results in the manuscript).
Round 2
Reviewer 1 Report
nutrients-830632-peer-review-v2
Overall feedback: It appears that my feedback from the last two rounds of review have not been fully taken into account or we rebutted in this revised version and cover letter. Some specific comments are found below.
From the abstract, it's still not clear to me that visceral adipose tissue volume divided by height cubed is a stronger indicator of metabolic risk compared to visceral adipose tissue volume alone. More statistics need to be shown in the abstract.
Minor point: if abbreviations are used in the manuscript, they should be used consistently. For example, the abbreviation for magnetic resonance imaging is different in the abstract from in the introduction. Also, the abbreviation does not seem to be defined in the abstract.
A Fisher’s z-transformation should be used to determine if the correlation coefficients between each of the relationships were significantly different from each other.
Information in the paper about whether visceral adipose tissue volume is a better predictor of metabolic health compared to simple anthropometric values is not original and unclear how this information adds to the literature. Similarly, information about the differences between men and women is not original and it is unclear how this knowledge advances the literature.
The information about vat normalized for body volume being a better predictor of metabolic health then VAT alone is the most interesting and useful part of the paper, but this part of the paper is not clearly explained. Moreover, as the relationship is only significant in one sex and not the other, at least for some parameters, then the conclusions to the study are over stated.
Reviewer 3 Report
Thank you for the effort to update your manuscript once again. Although I personally think the analyses and conclusions could be done differently, the paper has certainly improved with all the revisions you provided.
Keeping all else the same, I would double check some of the (new) results you included in Table 2 and how you reference them in the text of the results section:
- On the left-hand side of the table I think you should remove the column with "VAT vs ISIMats" and "VAT vs HbA1c" labels or replace it with just "ISIMats" and "HbA1c" as you are now evaluating the explained variance (R2) of all the separate factors in relation to ISIMats and HbA1c and not just VAT vs ISIMats or HbA1c.
- Secondly, the R2 reported for the association between age and HbA1c seems unnaturally high in TDFS in both men and women, especially compared with the results from the UKBB directly below them (almost 4-6 fold difference).
- In the text on line 335 the correlation between VAT-indices and HbA1c in the UKBB is said to lie between R2 of 0.013-0.044. The lowest I could find were 0.014 for %VAT in men and the highest 0.046 for VAT/m3 in women.
Author Response
Please see the attachment.

This manuscript is a resubmission of an earlier submission. The following is a list of the peer review reports and author responses from that submission.
Round 1
Reviewer 1 Report
In this research, the authors aimed at identifying best predictors of insulin sensitivity and glucose metabolism among indexes calculated on the bases of MRI-measured VAT volume in two large cohorts of individuals. Indeed, they found that VATi3 was the best correlate of insulin sensitivity, HbA1c levels and prediabetes/diabetes for the first cohort and HbA1c and prediabetes/diabetes in the second one. This research is interesting and well conducted, the rationale is strong since it is easy to understand that VAT volume alone cannot really reflect body shape and composition. Therefore, I see clinical impact of these results, in consideration of the high prevalence of obesity and its associated complications. Another merit of this study is the accurate metabolic phenotyping performed also by OGTT, which allowed to calculate static and dynamic indexes of insulin sensitivity in one of the two cohorts.
However, there are some point to be addressed.
Abstract:
“VATi3 most strongly correlated with insulin sensitivity, HbA1c levels and prediabetes/diabetes in the TDFS (0.277<r²<0.441, 0.744<AUC<0.854 for insulin sensitivity, 0.001<r²<0.090 for HbA1c levels and 0.648<AUC<0.743 for prediabetes) and HbA1c and prediabetes/diabetes (0.013<r²<0.044 and 0.609<AUC<0.692) in the UKBB for females and males in all age groups” the direction of these correlations is not clear. It seems that VATi3 is correlated both with (higher) insulin sensitivity and diabetes. Please, check this sentence.
Introduction
The authors affirm that the rationale behind focusing on height instead of weight is that there is evidence suggesting that height may be a protective factor, since taller individuals are less susceptible to metabolic diseases. However, even if I see the point, there is also a balk of literature on weight and CVD, so I think that this statement should be reformulated. Indeed, VAT volume itself gives approximately an idea of body weight, and including the height in the index, provide information on the body habitus, adding important information on the body shape. Consider to revise this paragraph, since it represents the rationale for the study design.
Methods
While it is well described that UKBB Cohort was recruited for a bigger study, it is less evident if TDFS Cohort has been recruited for this study. Please provide a symmetric description of the recruitment setting for the two cohorts.
Results
I suggest to format the tables in a fashion easier to understand. I would also suggest to present some of them as Supplementary data.
Line 336 “HbA1c level, as a marker for permanently elevated blood glucose” substitute permanently with chronically.
Reviewer 2 Report
In this manuscript, the predictive value of several VAT and VAT-related indices for insulin sensitivity, HbA1c levels and (pre)diabetes was examined in the Tübingen Diabetes Family Study (TDFS) and the UK Biobank. In both these studies, VAT was precisely measured using MRI, and in TDFS an oral glucose tolerance test (OGTT) was performed.
With the use of these precise measures in two large cohorts, this study has a lot of potential. However, I think the overall aim of this study should be more clear, and according to this aim the analyses could be changed to fit this aim.
My questions and suggestions to the authors are listed below.
- As I mentioned, the overall aim of this study is not clear for me. The aim as stated, is to identify the best predictor of insulin resistance and prediabetes or diabetes among several VAT indices. This raises two questions for me:
- What is the predictive performance of other prediction models for insulin resistance and (pre)diabetes? How well is the predictive performance of these models compared with the predictive performance of the models examined in this study? It would be more interesting and relevant to see these comparisons.
- In general, VAT indices obtained by MRI or 1H-MRS are not feasible in clinical practice when used in a prediction model. What will be the clinical implications of the author's findings?
- Were there contra-indications for undergoing MRI in both TDFS and UK Biobank? (I think of a certain maximum waist circumference, not having an ICD). Could this have influenced the results?
- Please mention the exact p-values instead of ‘n.s.’ when the p-values are above the p<0.05 threshold.
- In Table 4, it would be more informative to report the 95% confidence intervals of the r2, and AUC instead of the p-values.
- Do the authors have information on menopausal status in the TDFS and/or UK Biobank? It would be interesting to see the proportions of pre/peri/postmenopausal women in both studies.
- In the final conclusions, it is stated that ‘the proposed normative VAT index which volumetrically corrects for body height VATi3 has shown to be a slightly better predictor for insulin resistance and prediabetes as simply quantifying the VAT volume and its use is advised in studies for metabolic research’. I think in this case ‘metabolic research’ should be more specified e.g. predictive research, as the results of this study do not indicate that VATi3 is also a good variable to use in aetiological research.
Reviewer 3 Report
This is potentially interesting and important research, but the depth of analysis and interpretation is too shallow.
MAJOR
- The analysis of data seems overly simplistic. By collapsing age (a continuous variable) into discrete age groups, a lot of the richness of the data is lost. It would be better to do analyses where age can remain as a continuous variable. Also, it is not clear why the data was separated into males and females (and different age categories): a more sophisticated analysis model would enable the Authors to include sex and age in a single model of analysis, and this would be more powerful.
- It is not clear how the significant correlation in Figure 2 adds to the paper. Parameters can be highly correlated with each other without one parameter having diagnostic power of the other.
- The paper uses a lot of space to describe basic summary statistics of sub-groups separated by sex and age, but these tables and data seem irrelevant. What is relevant is to see whether another parameter besides VAT per se can be a better predictor of metabolic dysfunction.
- Figure 3 needs more explanation, as it is a new type of graph that myself (and likely other readers as obesity researchers) would not be familiar with.
- The different correlation coefficients of VAT, VAT/m, VAT/m2 and VAT/m3 are only marginally different from each other. Have the Authors done a test for significance of difference between those correlations? I doubt there would be a significant difference between the correlations, and that then calls into question the Authors’ conclusion that VAT/m3 is the best predictor of metabolic dysfunction.
- Correlations are not the same as diagnostic tools. A terrible diagnostic can have terrific correlation with the outcome.
- The green and black lines in Figure 4 look pretty similar to me. What is the AUC for these curves, and what is the point of optimum sensitivity and specificity for each? This data is not shown.
- The Abstract is exceedingly difficult to read. The great number of abbreviations contributes to the difficulty in following the Abstract.
- The whole manuscript uses a large number of abbreviations, and that makes it overly difficult to read. It would be better to reduce abbreviations as much as possible.
- Some of the abbreviations could be made simpler to read by giving them intuitive names. For example, instead of VATi2, it could be better to write VAT/m2, akin to how BMI units are written in kg/m2.
- The paper needs to use person-first language around obesity throughout. For example, instead of saying 100% of Americans will BE overweight or obese, it is more correct now to say 100% of Americans will HAVE overweight or obesity (or a BMI in the overweight or obese range).
Reviewer 4 Report
The authors investigated the association between several MRI-derived measures of adiposity, in particular visceral adipose tissue (VAT) indices constructed from VAT and other measures such as height, and measures of insulin sensitivity/resistance and altered glucose metabolism defined as impaired fasting glycemia or diabetes.
The overall conclusion is that a measure of VAT divided by height cubed is most strongly associated with insulin resistance and glucose metabolism status and most predictive of insulin resistance.
Although the authors performed an extensive number of analyses, I think in its currents form the manuscript suffers from a couple of methodological/statistical problems and the conclusions are not fully supported by the results that are presented. In addition, the manuscript does not add much new information to the existing knowledge base of this topic as associations of age, sex and insulin/glucose metabolism with VAT (and VAT indices) have been described previously.
My primary concern is the setup of the prediction component in this study. Both the outcomes and predicting variables were cross-sectionally assessed at the same moment in time. As such, what is tested here is the diagnostic performance rather than the (prospective) predictive performance of these indices for incident disease. Since detecting currently present insulin resistance and/or impaired glucose metabolism can be easily done through blood testing it is unclear what the benefit of using VAT or VAT-indices for this would be. A similar analysis with incident disease might be more informative. In addition, only discrimination assessed with the AUC has been evaluated, while for prediction it is also recommended to evaluate, amongst others, measures of calibration. The AUC is also well known for being reasonably insensitive to changes in predictive performance, so judging and comparing predictive performance based on the AUC alone is advised against. If it is not truly the intent to assess the predictive performance of these indices but only to evaluate which of the indices associates most strongly with for example insulin resistance, then I would recommend omitting the ROC analyses and all references to prediction and focus instead on being a descriptive study of these associations (i.e. just describe the associations and don’t make any claims about the predictive or causal role of these associations).
A second concern is the interpretation of the values from the different analyses in the second part of the manuscript (page 11 onwards). Although it is true that the various outcome measures (R2, AUC) presented in Table and Figure 4 are, in general, the highest for VATi3, the actual differences between the indices are for the most part small (increase in R2 is mostly around 2-3% and the increase in AUC mostly 0.01-0.02). These differences are likely for the most part inconsequential at least for the associations with the outcomes that are evaluated here and seem to indicate that there are only small differences between height- and non-height adjusted measures of VAT.
Outside of these main issues, there are some smaller issues that could be addressed to improve the analyses:
- For the comparisons between men and women in tables 1 and 2 it would be more informative to present mean differences with confidence intervals rather than p-values.
- For the analyses reported in Table 3: unless you expect strong non-linear associations, the stratification into different age groups could be foregone and replaced with linear regressions with VAT(-indices) as outcome and age, IR/DIAB-status and an interaction term of the both as independent variables. The results could be described more concisely than these tables.
Minor points/remarks:
The descriptive texts for some of the figures and tables could be clarified/updated.
Figure 3: what do the red circles represent in the context of this figure? It is not explained what they represent and overall decrease the readability of the figure.
Figure 4: The caption below the figure refers to Table 3, however the information is summarized in Table 4 in this version of the manuscript. Caption could also include the outcome for which these analyses were performed which can now only be found in the main text. Additionally, what do the straight diagonal lines in the graphs represent?
Throughout the manuscript, R2 (explained variance) is frequently referred to as correlation coefficients. Although in the context of a univariable regression analysis the explained variance (R2 and correlation coefficient (r) have a one-on-one relation, it is still incorrect to refer to the R2 as a correlation. It is also not completely clear if the R2 were calculated using the continuous ISI and HbA1c or the dichotomized versions; please clarify this in the methods section.